# The Anti-Obesity and Anti-Steatotic Effects of Chrysin in a Rat Model of Obesity Mediated through Modulating the Hepatic AMPK/mTOR/lipogenesis Pathways

**DOI:** 10.3390/molecules28041734

**Published:** 2023-02-11

**Authors:** Ghaleb Oriquat, Inas M. Masoud, Maher A. Kamel, Hebatallah Mohammed Aboudeya, Marwa B. Bakir, Sara A. Shaker

**Affiliations:** 1Department of Medical Laboratory Sciences, Faculty of Allied Medical Sciences, Al-Ahliyya Amman University, Amman 19328, Jordan; 2Department of Pharmaceutical Chemistry, Faculty of Pharmacy, Pharos University in Alexandria, Alexandria 21311, Egypt; 3Department of Biochemistry, Medical Research Institute, Alexandria University, Alexandria 21561, Egypt; 4Department of Human Physiology, Medical Research Institute, Alexandria University, Alexandria 21561, Egypt; 5Department of Pharmacology and Experimental Therapeutics, Alexandria University, Alexandria 21561, Egypt

**Keywords:** non-alcoholic fatty liver disease, flavonoids, chrysin, AMP-activated protein kinase (AMPK), TOR Serine-Threonine Kinases (mTOR), mitochondrial biogenesis

## Abstract

Background: Obesity is a complex multifactorial disease characterized by excessive adiposity, and is linked to an increased risk of nonalcoholic fatty liver disease (NAFLD). Flavonoids are natural polyphenolic compounds that exert interesting pharmacological effects as antioxidant, anti-inflammatory, and lipid-lowering agents. In the present study, we investigated the possible therapeutic effects of the flavonoid chrysin on obesity and NAFLD in rats, and the role of AMP-activated protein kinase (AMPK)/mammalian target of rapamycin (mTOR) pathways in mediating these effects. Method: Thirty-two Wistar male rats were divided into two groups: the control group and the obese group. Obesity was induced by feeding with an obesogenic diet for 3 months. The obese rats were subdivided into four subgroups, comprising an untreated group, and three groups treated orally with different doses of chrysin (25, 50, and 75 mg/kg/day for one month). Results revealed that chrysin treatment markedly ameliorated the histological changes and significantly and dose-dependently reduced the weight gain, hyperglycemia, and insulin resistance in the obese rats. Chrysin, besides its antioxidant boosting effects (increased GSH and decreased malondialdehyde), activated the AMPK pathway and suppressed the mTOR and lipogenic pathways, and stimulated expression of the genes controlling mitochondrial biogenesis in the hepatic tissues in a dose-dependent manner. In conclusion, chrysin could be a promising candidate for the treatment of obesity and associated NAFLD, aiding in attenuating weight gain and ameliorating glucose and lipid homeostasis and adipokines, boosting the hepatic mitochondrial biogenesis, and modulating AMPK/mTOR/SREBP-1c signaling pathways.

## 1. Introduction

Obesity is a pandemic concern due to its increasing prevalence around the world [1]. Obesity is a gateway for other comorbidities, such as metabolic syndrome, cardiovascular diseases, and insulin resistance, which are the main risk factors for the development of non-alcoholic fatty liver diseases (NAFLD) [2]. NAFLD is defined as the accumulation of fat in the liver of patients who do not consume excessive alcohol. It manifests itself as simple steatosis, which means lipid accumulation in the hepatic tissue, and over time progresses to nonalcoholic steatohepatitis (NASH), cirrhosis, and hepatocellular carcinoma [3]. Obesity plays an important role in both the initial process leading to simple steatosis and its progression to NASH [4]. Obesity may affect the liver through different interrelated hepatic pathways that may include, adipokines, AMPK, mTOR, lipogenesis, and mitochondrial homeostasis.

The adipokines (e.g., leptin, adiponectin, and tumor necrosis factor-α (TNF-α)) derived from the adipose tissue, may contribute to simple steatosis and NASH [5,6]. The inhibition of AMPK or its upstream activator, liver kinase B1 (LKB1), is associated with obesity [7]. AMPK functions as a metabolic sensor to monitor cellular energy status by at least three mechanisms: suppression of hepatic de novo lipogenesis, increased fatty acid oxidation in the liver, and promotion of mitochondrial function [8]. AMPK downregulates the expression and activation of sterol regulatory element binding protein-1c (SREBP-1c), which is the master transcriptional regulator of lipogenesis [9]. The obese patients showed reduced AMPK activity in adipose tissues, and this reduction is correlated with whole-body insulin resistance, suggesting the important role of AMPK in obesity [10,11,12]. Obesity is associated with the induced mTOR pathway, which is implicated in many comorbidities, including NAFLD. In the liver, mTORC1 acts as a promotor of SREBP-dependent lipogenesis and inhibits lipophagy by blocking autophagy initiation and attenuating lysosome biogenesis [13]. The mTORC1 pathway was reported to be activated in the hepatic tissues of HFD-rats, which causes the accumulation of misfolded or unfolded proteins, aggravates endoplasmic reticulum stress [14], and contributes to the induction of de novo lipogenesis, which induces NAFLD pathogenesis and accelerates NAFLD-related hepatocellular carcinoma development [15]. Additionally, the dysregulation of mitochondrial homeostasis is linked to obesity [16]. Mitochondrial biogenesis is essential to augment mitochondrial capacity, which helps relieve lipid accumulation in the liver. Proliferator-activated receptor γ coactivator-1α (PGC-1α) is a key regulator of energy homeostasis by transcriptional regulation of genes involved in fatty acid oxidation and mitochondrial biology. Previous research has shown a 40% decrease in hepatic PGC-1α expression in NAFLD patients, accompanied by mitochondrial dysfunction, lipid accumulation, and insulin resistance [17].

The search for a rapid way to treat obesity and associated metabolic derangements is mandatory. The current guidelines advise medication as the second line of treatment for obesity after lifestyle modifications, but most lifestyle approaches lead to gradual weight loss over six months, followed by a plateau and weight gain over one to three years [18]. Therefore, anti-obesity treatment should be considered as part of a comprehensive strategy for the treatment of obesity and related comorbidity like NAFLD. Such treatments should target the pathway involved in the pathogenesis and progression of the disease. Currently, there are no approved drugs for the treatment of NAFLD. Several drugs, such as pioglitazone, vitamin E, and pentoxifylline have been screened, but they have shown limited benefits [19]. There is an increasing focus on nutraceuticals for the treatment of metabolic disorders, including obesity and associated hepatic manifestations.

The intake of flavonoids has been found to protect against NAFLD [20]. Chrysin is one of the major flavonoids found in plants belonging to the genus Passiflora and is also found in bee propolis. It has been reported to have numerous pharmacological activities, including neuroprotective, antidiabetic, anticancer, nephroprotective, cardioprotective, antiarthritic, and anti-asthmatic effects. Chrysin has hepatoprotective effects against hepatotoxins, such as ethanol and carbon tetrachloride, has hypolipidemic and anti-inflammatory activities, and ameliorates diabetes-induced liver damage in rats and mice. Pai and his colleagues found that chrysin exerted an anti-obesity effect by inhibiting the pancreatic lipase, reducing sucrose preference, reducing calorie intake, and increasing the locomotor activity of rats [20,21]. However, the metabolic and hepatic effects of chrysin have not been studied in detail. Therefore, we performed this study to investigate the anti-obesity and anti-steatotic effects of chrysin in an HFD-rat model of obesity and to explore the possible mechanism(s) of these actions through the study of the changes in the pathogenic pathways involved in the hepatic manifestations of obesity, including glucose and lipid homeostasis, AMPK/mTOR/lipogenesis pathways, adipokines, redox status, inflammation, and mitochondrial biogenesis.

## 2. Results

### 2.1. Body Weight Change during Treatment Period

The obese rats showed significantly higher weight (initial, final weight, and weight gain) as compared with control rats (*p* < 0.001). The obese rats treated with chrysin showed significantly lower final weight and weight gain compared with untreated obese rats in a dose-dependent manner (*p* < 0.001), although the final weight and weight gain of treated rats was still significantly higher than the control rats (Table 1).

### 2.2. Glucose Homeostasis Parameters

Untreated obese rats showed significantly higher fasting blood sugar and insulin levels compared with the control rats (*p* < 0.001), while the obese rats treated with chrysin showed significantly lower values of FBG and insulin compared with the untreated obese rats in a dose-dependent manner (*p* < 0.001) (Table 2). The insulin resistance index calculated by the HOMA model (HOMA-IR) indicated that the untreated obese rats had significantly higher HOMA-IR compared with the control rats (*p* < 0.001), while the obese rats treated with chrysin had a significantly lower value of HOMA-IR compared with the untreated obese rats in a dose-dependent manner (*p* < 0.001). Although the levels of fasting insulin and HOMA-IR were significantly lower in treated obese rats compared with untreated obese rats, their levels were still higher in treated obese rats compared with the control group (Table 2).

### 2.3. Serum Lipids Profile and Hepatic Triglyceride Content

The untreated obese rats had significantly higher levels of serum TG, TC, and LDL-C, while showing a significantly lower level of HDL-C compared with the control rats (*p* < 0.001). The treatment of the obese rats with chrysin resulted in a significant dose-dependent decline in the levels of TG, TC, and LDL-C and a dose-dependent increase in HDL-C compared with the untreated obese rats (*p* < 0.001) (Table 2). The chrysin treatment at the doses of 50 and 75 mg/kg completely normalized the levels of the lipid profile components. Regarding the hepatic TG content, the untreated obese rats showed significantly higher levels compared with control rats (*p* < 0.001) while the obese rats treated with chrysin showed a significant dose-dependent decline compared with the untreated obese rats (*p* < 0.001) (Table 2).

### 2.4. Liver Function Tests

Untreated obese rats had significantly higher serum activities of ALT and AST and higher serum levels of bilirubin compared with control rats (*p* < 0.001). The obese rats treated with chrysin showed significantly lower values compared with the untreated obese rats in a dose-dependent manner (*p* < 0.001) (Table 2).

### 2.5. Adipocytokines Levels

The untreated obese rats showed significantly higher levels of serum TNF-α and leptin compared with control rats (*p* < 0.001), treating the obese rats with chrysin significantly decreased the levels of TNF-α and leptin compared with untreated obese rats in a dose-dependent manner (*p* < 0.001) (Table 3). The adiponectin level showed the opposite pattern of change, as it significantly decreased in the untreated obese rats compared with the control group (*p* < 0.001), while it significantly and dose-dependently increased in the obese rats treated with chrysin compared with the untreated obese rats (*p* < 0.001) (Table 4).

### 2.6. Hepatic Redox Parameters

Hepatic MDA content and the GSH system (tGSH, GSH, GSSG, and GSH/GSSG ratio) are presented in Table 5. The untreated obese rats had a significantly higher level of MDA and oxidized glutathione (GSSG) compared with control rats (*p* < 0.001), while the obese rats treated with chrysin showed a significant dose-depended decline in their values compared with untreated obese rats (*p* < 0.001). On the other hand, the hepatic levels of total and reduced GSH were significantly lower in the untreated obese rats compared with the control group (*p* < 0.001), while they were significantly and dose-dependently increased in the rats treated with chrysin compared with the untreated obese rats (*p* < 0.001). Regarding the redox ratio (GSH/GSSG, the untreated obese rats had a marked decline in the ratio compared with the control rats, and the obese rats treated with chrysin showed a significantly higher value than untreated obese rats in a dose-dependent manner (*p* < 0.001) (Table 4).

### 2.7. Hepatic Expression of AMPK at mRNA and Protein Levels

The obese rats had significantly suppressed hepatic expression of AMPK at mRNA, lower by about 50% than the healthy control rats (*p* < 0.001). At the protein level, the obese rats had a significantly lower level of active phosphorylated AMPK (p-AMPK), lower by about 34% than the control rats (*p* < 0.001) (Figure 1A). All the obese rats treated with chrysin showed significantly upregulated expression of AMPK at mRNA level compared with the untreated rats, with no significant differences from the control rats and no significant differences between the different doses used (*p* < 0.001). The level of the active phosphorylated form of AMPK showed a significant dose-dependent increase in the hepatic tissues of obese rats treated with chrysin compared with the untreated obese rats. The chrysin treatment at the highest dose (75 mg/Kg) completely normalized the hepatic level of phosphorylated AMPK protein (*p* < 0.001) (Figure 1B).

### 2.8. Hepatic Expression of LKB 1 at mRNA and Protein Levels

The obese rats had significantly suppressed hepatic expression of LKB 1 at mRNA, lower by about 34% than the healthy control rats (*p* = 0.001). The active LKB 1 protein level in obese rats was lower by about 38% than the control rats (*p* < 0.001) (Figure 2A). Obese rats treated with chrysin doses (50 mg/kg & 75 mg/kg) showed significantly upregulated expression of LKB1 at mRNA level compared with the untreated rats, with no significant differences from the control rats (*p* < 0.001). The level of LKB1 protein showed a significant dose-dependent increase in the hepatic tissues of obese rats treated with chrysin compared with the untreated obese rats (*p* < 0.001). The chrysin treatment at the highest dose (75 mg/Kg) completely normalized the hepatic level of LKB1 protein (Figure 2B).

### 2.9. Hepatic Expression of SREBP-1c at mRNA and Protein Levels

Hepatic expression of SREBP-1c at mRNA was significantly elevated in obese rats, about three times higher than healthy control rats (*p* < 0.001). However, the active SREBP-1c protein level in obese rats was significantly decreased by about 38% compared with the control rats (*p* < 0.001) (Figure 3A). Obese rats treated with chrysin doses showed significantly lower expression of SREBP-1c at mRNA level in a dose-dependent manner compared with untreated rats (*p* < 0.001). However, obese rats treated with chrysin doses (50 mg/kg & 75 mg/kg) showed a significant elevation of active SREBP-1c protein level compared with the untreated rats, with significant differences from the control rats (*p* < 0.001) (Figure 3B).

### 2.10. Hepatic Expression of PGC-1a at mRNA and Protein Levels

The obese rats had significantly suppressed hepatic expression of PGC-1a at mRNA, lower by about 64% than non-obese control rats (*p* < 0.001). In addition, the PGC-1 protein level in obese rats was lower by about 35% than the control rats (*p* < 0.001) (Figure 4A). Only obese rats treated with chrysin at the highest dose (75 mg/Kg) had significantly upregulated expression of PGC-1a at mRNA level compared with the untreated rats (*p* < 0.001), with no significant differences from the control rats. The level of PGC-1 a protein showed a significant dose-dependent increase in the hepatic tissues of obese rats treated with chrysin compared with the untreated obese rats (*p* < 0.001) (Figure 4B).

### 2.11. Hepatic Expression of NRF 1 at mRNA and Protein Levels

The obese rats had significantly suppressed hepatic expression of NRF 1 at mRNA, lower by about 21% than non-obese control rats (*p* < 0.001). At the NRF1 protein level, the obese rats had a significantly lower level, about 36% lower than the control rats (*p* < 0.001) (Figure 5A). Only obese rats treated with chrysin at the highest dose (75 mg/Kg) significantly upregulated expression of NRF 1 at mRNA level compared with the untreated rats (*p* = 0.007) with no significant differences from the control rats. The level of NRF-1 protein in obese rats treated with chrysin doses (50 mg/kg & 75 mg/kg) showed significant elevation compared with the untreated rats (*p* < 0.001) (Figure 5B).

### 2.12. Hepatic Expression of Tfam at mRNA and Protein Levels

The obese rats had significantly suppressed hepatic expression of Tfam at mRNA by about 21% compared with non-obese control rats (*p* < 0.001). At the Tfam protein level, the obese rats had a significantly lower level by about 53% compared with the control rats (*p* < 0.001) (Figure 6A). All the obese rats treated with chrysin showed no significant difference in the expression of Tfam at the mRNA level compared with the untreated obese rats. The level of Tfam protein level in obese rats treated with chrysin doses (50 mg/kg & 75 mg/kg) showed a significant elevation in its level compared with the untreated obese rats (*p* = 0.001, *p* < 0.001, respectively) (Figure 6B).

### 2.13. Hepatic Expression of mTOR at mRNA and Protein Levels

Hepatic expression of mTOR at mRNA was significantly elevated in obese rats, double that of non-obese control rats (*p* < 0.001). However, mTOR protein expression in obese rats was 1.6-fold higher than the control rats (*p* < 0.001) (Figure 7A). Obese rats treated with chrysin doses showed significantly lower expression of mTOR at mRNA level in a dose-dependent manner compared with untreated rats. Moreover, obese rats treated with chrysin doses (50 mg/kg & 75 mg/kg) showed a significant decline in mTOR protein level compared with the untreated rats (*p* < 0.001) (Figure 7B).

### 2.14. Histopathological Examinations of Hepatic Tissues

The hepatic sections from normal rats exhibited normal architecture with plates of hepatocytes radiating from the central vein. Most of the hepatocytes were mononucleated and in contact with blood sinusoids, and Kupffer cells were associated with sinusoidal lining cells (Figure 8A). The hepatic sections from the obese rats showed degenerated hepatic architecture with congested central and portal veins, increased inflammatory cell infiltration, dilated sinusoids, fatty degeneration, macrosteatosis, and ballooned deeply eosinophilic cytoplasm (Figure 8B,C). Treatment of obese rats with increasing doses of chrysin resulted in dose-dependent hepatic histological improvements (Figure 8D–F).

## 3. Materials and Methods

### 3.1. Materials

Chrysin was purchased from Sigma-Aldrich (St. Louis, MO, USA). A variety of colorimetric kits were obtained from (Bio-Med Diagnostic Inc., White City, OR, USA) for the assay of serum levels of fasting blood glucose (FBG), alanine aminotransferase (ALT), aspartate aminotransferase (AST), bilirubin, triglycerides (TG), total cholesterol (TC), and high-density lipoprotein-cholesterol (HDL-C). ELISA kits for the assay of serum leptin were assayed using a rat ELISA kit (eBioscience, San Diego, CA, USA), adiponectin was assayed using a rat ELISA kit (Elabscience, Houston, TX, USA),and insulin and TNF-α were assayed using rat-specific ELISA kits (Chongqing Biospes Co., Ltd., Chongqing, China, catalog no. BEK1243, and BEK1214, respectively). The hepatic supernatant was used for the determination of phosphorylated-AMPK at Thr172 (P-AMPK), LKB1, and mTOR proteins using a rat-specific ELISA kit (LSBio, Seattle, WA, USA, Cat. No. LS-F36060, LS-F13457, and LS-F17553; respectively), according to the manufacturer’s instructions. Additionally, PGC-1α and mitochondrial transcription factor A (Tfam) proteins were assayed using a rat-specific ELISA kit (MyBioSource, San Diego, CA, USA, Cat No. MBS2706379, and MBS1600609), while nuclear respiratory factor 1 (NRF-1) and SREBP-1c proteins were assayed using rat-specific ELISA kits (Chongqing Biospes Co., Ltd., China, Cat. No. BYEK2318 and BYEK3082).

### 3.2. Experimental Animals

The study was conducted on 2-month-old 32 male Wister rats. Rats were obtained from the animal house facility in Medical Technology Center, Medical Research Institute, Alexandria University, Egypt. Animals were kept 5 per cage at 23 °C in a 12 h light/12 h dark cycle under good hygienic conditions and standard humidity with access to food and water.

### 3.3. Ethical Statement

The study was approved by the Institutional Animal Care and Use Committee (IACUC)-Alexandria University, Egypt (AU0122232231). All steps were performed following the guidelines for the care and use of laboratory animals (USA National Institute of Health Publication No 80-23, revised 1996), and all efforts were made to reduce the distress of rats during the whole experimental period.

### 3.4. Induction of Obesity

Obesity was induced in young male Wister rats (about 2 months of age and weighing 120–140 g) by feeding them with an obesogenic diet for 3 months. Rats that were 20% heavier than the mean weight of the control rats of the same age were considered obese. The composition of the obesogenic diet used in this experiment was: 30 g protein (300 kcal), 26.5 g fat (195 kcal lard, 70 kcal corn oil), 36.5 g carbohydrate (105 cal dextran, 106 cal corn starch, 140 kcal sucrose), 3 g vitamin mix (30 kcal), and 4 g mineral mix (40 kcal) per 100 g diet [22].

### 3.5. Experimental Design

Animals were classified into two main groups. Group I (control group): 8 rats were fed with a normal chow containing 22.2 g of protein, 63.1 g of carbohydrate, 4.3 g of fat (oil), 5.4 g of fiber, 1 g of vitamins, and 4 g of minerals per 100 g diet for 4 months. Group II (obesity group): after the establishment of obesity, the 32 obese male rats were fed with an obesogenic diet and subdivided into four groups (eight rats each) according to the treatment. GIIA: untreated obese male rats. GIIB: obese rats treated with chrysin at a dose of 25 mg/kg daily by oral gavage for one month. GIIC: obese rats treated with chrysin at a dose of 50 mg/kg daily by oral gavage for one month. GIID: obese rats treated with chrysin at a dose of 75 mg/kg daily by oral gavage for one month. The doses were selected according to the study by Pai et al., 2020 [20,21].

### 3.6. Collection of Samples

At the end of the treatment period, the rats were fasted overnight. Then, all rats were anesthetized by intraperitoneal injection of ketamine (75 mg/kg) and xylazine (10 mg/kg) and then sacrificed by cervical dislocation. The serum samples were prepared by collecting the blood from the retroorbital vein in anticoagulant-free tubes followed by centrifugation at 3000× *g* for 10 min. The serum samples were used for the determination of FBG, insulin, homeostasis model assessment index for insulin resistance (HOMA-IR), lipid profile (TG, TC, HDL-C, low-density lipoprotein cholesterol (LDL-C)), ALT activity, AST activity, leptin, adiponectin, and TNF-α levels. The tissue samples of the liver were taken from the same region in each group, washed with cold saline solution, weighed, and divided into three aliquots. The first one was immersed in RNA later solution, stored at −80 °C, and used for total RNA isolation for the assessment of gene expression. The second was used for lipid extraction for the determination of the lipid content of the liver. The last aliquot was homogenized in RIPA buffer in a ratio of 1:9 and centrifuged at 10,000× *g* for 10 min at 4 °C, and the supernatant was stored in aliquots for subsequent determinations of total protein level by Lowry method, triglyceride content, malondialdehyde (MDA) as an index of lipid peroxidation, and glutathione (GSH), and the protein content of p-AMPK Thr172, p-LKB1, PGC-1α, Tfam, NRF-, SREB1C, and mTOR by ELISA.

### 3.7. Serum Parameters

Serum levels of fasting blood glucose (FBG), alanine aminotransferase (ALT), aspartate aminotransferase (AST), bilirubin, triglycerides (TG), total cholesterol (TC), and high-density lipoprotein-cholesterol (HDL-C) were assayed using commercially available kits (Bio-Med Diagnostic Inc., USA). Low-density lipoprotein-cholesterol (LDL-C) was estimated according to Friedewald’s equation, LDL-C (mg/dl) = TC − (HDL-C) − (TG/5).

Serum leptin was assayed using a rat ELISA kit (eBioscience, San Diego, CA, USA), adiponectin was assayed using a rat ELISA kit (Elabscience, Houston, TX, USA), and insulin and TNF-α were assayed using rat-specific ELISA kits (Chongqing Biospes Co., Ltd., catalog no. BEK1243, and BEK1214, respectively). All procedures were performed according to the manufacturer’s instructions. The HOMA-IR was then calculated using the following formula [23]:HOMA-IR = (Fasting insulin((µIU)/mL) × Fasting glucose(mg/dL))/(22.5 × 18)

### 3.8. Determination of Hepatic Protein Contents of p-AMPK Thr172, LKB1, PGC-1α, Tfam, NRF1, SREBP-1c and mTOR Using ELISA

The hepatic supernatant was used for the determination of phosphorylated-AMPK at Thr172 (P-AMPK), LKB1, and mTOR proteins using rat-specific ELISA kit (LSBio, Seattle, Washington, DC, USA, Cat. No. LS-F36060, LS-F13457, and LS-F17553; respectively), according to the manufacturer instructions. Additionally, PGC-1α and Tfam proteins were assayed using a rat-specific ELISA kit (MyBiosource, San Diego, CA, USA, Cat No. MBS2706379, and MBS1600609), while NRF1 and SREBP-1c proteins were assayed using rat-specific ELISA kits (Chongqing Biospes Co., Chongqing, China, Cat. No. BYEK2318 and BYEK3082) according to the instructions of the manufacturers. The total protein concentration was determined using Lowry’s method [24].

### 3.9. Tissues Contents of Total Reduced and Oxidized Glutathione

Glutathione (GSH) and glutathione disulfide (GSSG) were assayed using the method of Griffith, which depends on the oxidation of GSH by 5,5′-dithiobis-(2-nitrobenzoic acid) (DTNB) to yield GSSG and 5-thio-2-nitrobenzoic acid (TNB). Oxidized GSSG is reduced enzymatically by the action of glutathione reductase and NADPH to regenerate GSH. The rate of TNB formation is monitored at 412 nm and is proportional to the sum of GSH and GSSG present in the sample [25]. Liver samples were immediately homogenized in metaphosphoric acid (0.5 gm liver: 4.5 mL metaphosphoric acid). The homogenates were centrifuged at 10,000 rpm for 10 min at 40 °C. The serum samples were immediately precipitated by precipitant reagent (0.2 M metaphosphoric acid, 0.5 M NaCl. metaphosphoric acid, 0.6 mM EDTA) (1:1 *v*/*v*), and centrifuged at 10,000 rpm for 10 min at 40 °C. The supernatants were used for the determination of total and oxidized glutathione.

### 3.10. Determination of Malondialdehyde (MDA) as Thiobarbituric Acid Reactive Substances (TBARS)

Malondialdehyde was determined according to the method of Draper and Hadley. The tissue samples are heated with thiobarbituric acid (TBA) at low pH. The resulting pink chromogen has a maximal absorbance at 532 nm [26]. An aliquot of 0.1 mL of the sample was pipetted into a tube containing an equal volume of SDS solution. This was followed by the addition of 0.75 mL of acetic acid, 0.75 mL of TBA, and 0.3 mL of distilled water. The contents of the tubes were then mixed with a vortex. The tubes were incubated in a boiling water bath for 1 h and then cooled to room temperature. An aliquot of 0.5 mL of distilled water was added to each tube, followed by the addition of 2.5 mL n butanol. The contents of the tubes were vigorously mixed with a vortex and then rotated in a centrifuge at 4000 rpm for 10 min. The absorbance of the organic layer was read at 532 nm using a spectrophotometer against a blank containing phosphate buffer solution instead of the sample.

### 3.11. Determination of Triglyceride Content

Hepatic triglyceride contents were isolated from the hepatic tissues and determined using a minor modification of the Folich method [27]. An amount of 50 mg of liver tissue was homogenized in 5 mL of a chloroform/methanol (2:1) mixture. The extract was centrifuged at 2500× *g* for 15 min, and the supernatant was collected and evaporated to dryness under nitrogen. The residue was subsequently reconstituted in a solution of isopropyl alcohol containing 10% triton X and centrifuged at 10,000× *g* for 10 min. The supernatant was used for the determination of triglycerides using commercially available kits (Bio-Med Diagnostic Inc., White City, OR, USA).

### 3.12. Hepatic Expression of AMPK, LKB1, mTOR, PGC-1α, Tfam, NRF1, and SREB-1c

Thirty mg of the liver was used for total RNA extraction using the miRNeasy Mini Kit (Qiagen, Germany) according to the manufacturer’s instructions, and the concentration and integrity of the extracted RNA were checked using nanodrop. The reverse transcription of the extracted RNA was performed using reverse transcription (RT) was performed by TOPscript™ RT DryMIX kit (dT18/dN6 plus) (Enzynomics, Daejeon, Republic of Korea) according to the manufacturer’s instructions. The tissue expression of AMPK, LKB1, mTOR, PGC-1α, Tfam, NRF-1, and SREB1C was quantified in the cDNA by CFX Maestro™ Software (Bio-Rad, Hercules, CA, USA) using QuantiNova™ SYBR^®^ Green PCR Kit (Qiagen, Germany). Quantitative PCR amplification conditions were adjusted as an initial denaturation at 95 °C for 10 min and then 45 cycles of PCR for amplification as follows. Denaturation at 95 °C for 20 s, annealing at 55 °C for 20 s and extension at 70 °C for 15 s. The housekeeping gene 18S rRNA was used as a reference gene for normalization. The primers used for the determination of rat genes are presented in Table 5. The relative change in mRNA expression in the samples was estimated using the 2^−ΔΔCt^ method.

### 3.13. Histopathological Examination

The liver was isolated and fixed in 10% formalin, sectioned at 5 μm and stained with Harris hematoxylin and eosin stain (H&E) for histopathological examination.

### 3.14. Statistical Analysis

Data were analyzed using SPSS software package version 18.0 (SPSS, Chicago, IL, USA). The data were expressed as mean ± standard deviation (SD) and analyzed using one-way analysis of variance (ANOVA) to compare between different groups. The *p*-value was assumed to be significant at *p* < 0.05. The correlation coefficients (r) between different assayed parameters were evaluated using the Pearson correlation coefficient; *p* < 0.05 was considered as the significance limit for all comparisons.

## 4. Discussion

The flavonoid chrysin could be a promising candidate for the treatment of HFD-induced metabolic disturbances, as it significantly reduced the weight gain, corrected hyperglycemia, insulin resistance, and dyslipidemia, and improved adipocytokines in the HFD-obese rats in a dose-dependent manner. Additionally, in the liver, chrysin significantly ameliorated the HFD-induced derangements in metabolic pathways, mitochondrial biogenesis, and redox status.

The hepatic manifestations of obesity are clear in the present model as the obesogenic phenotype of the HFD-obese rats (marked weight gain, hyperglycemia, insulin resistance, dyslipidemia, elevation in the circulatory TNF-α and leptin levels and a marked decline in the adiponectin level) is associated with histological changes in the liver and significant elevation of serum activities of the transaminases and bilirubin level. These derangements can lead to the deregulation of the hepatic sterol response system, resulting in increased hepatic absorption and de novo synthesis of fatty acids and subsequent triglyceride synthesis, enhancing the steatosis of the liver [28]. Our results support this suggestion, as indicated by a marked accumulation of hepatic triglycerides and enhanced hepatic expression of SREBP-1c at both mRNA and protein levels in obese rats, which indicates steatosis of the liver. Previous reports documented that the upregulated hepatic expression of SREBP-1c by increased insulin causes selective induction of lipogenic genes. SREBP-1c, the only SREBP isoform induced by insulin, is transcriptionally induced at mRNA as well as the proteolytically activated by insulin, leading to the induction of hepatic lipogenesis pathway and the development of hepatic steatosis [29]. These metabolic abnormalities in the hepatic tissue of HFD-obese rats were associated with impairment of the glutathione antioxidant system (as indicated by a marked decline in total and reduced GSH and marked elevation of the oxidized form, GSSG) and induction of oxidative stress and lipid peroxidation, findings that agree with the study carried out by Parsanathan and Jain, who showed that the glutathione biosynthesis pathway activity was decreased in the livers of HFD-fed mice [30]. Many previous reports have documented such impairment of the hepatic GSH system in a rat model of obesity or NAFLD [31,32].

The HFD induced significant dysregulated expression of the important hepatic signaling pathways that regulate energy metabolism, including AMPK, mTOR, and mitochondrial biogenesis. The expression of AMPK and its upstream activating kinase (LKB1) was significantly suppressed, as well as p-AMPK and LKB1 protein levels in the liver, while the mTOR expression was markedly enhanced. AMPK and mTOR are mutually antagonistic nutrient sensors that have been associated with several metabolic diseases, such as NAFLD [33]. The phosphorylation of Tuberous sclerosis 2 (TSC2) and Raptor are both required for AMPK-mediated inhibition of mTORC1 [34]. mTORC1 was found to regulate lipid metabolism by phosphorylation of lipin-1, which allows nuclear SREBP-1c binding to lipogenic genes, resulting in the promotion of lipogenesis [35]. This hepatic impairment of the LKB1/AMPK pathway and induction of the mTOR/lipogenesis pathway in HFD-obese rats is coupled with significant suppression in the mitochondrial biogenesis pathway, as indicated by downregulation of the main components of the pathway, namely, PGC-1α, NRF1, and Tfam at both mRNA and protein levels. This pathway is essential for the maintenance of mitochondrial biogenesis and homeostasis, enhancement of mitochondrial capacity, and fatty acid oxidation [36]. In line with these data, Lei and his colleagues documented decreased hepatic expression of PGC-1α, NRF1, and Tfam in the NAFLD human liver, suggesting a key role in mitochondrial biogenesis [37]. Accumulating evidence suggests that NAFLD might be a mitochondrial disease, since mitochondrial dysfunction not only promotes ROS production and oxidative stress, but also affects hepatic lipid homeostasis and increases inflammation and cell death [38].

In the present study, our selection of chrysin as a candidate anti-obesity agent was driven by a growing body of evidence of its biological activities, including neuroprotective, antidiabetic, anticancer, nephroprotective, cardioprotective, antiarthritic, and anti-asthmatic effects [39,40,41], and by the fact that it is a part of the human diet, present in honey, propolis, and various medicinal plants and fruits, such as bitter melon and the peel of passion fruit [39]. Therefore, in this study, we assessed the anti-steatotic effects of chrysin in HFD-obese rats.

The treatment of obese rats with chrysin showed marked dose-dependent improvements in the hepatic tissue histology and was able to reverse cellular damage and protect cells from inflammation, apoptosis, and necrosis, which was associated with a significant reduction in the final weight and weight gain in a dose-dependent manner. The weight-lowering effect of chrysin was associated with prominent ameliorating effects on glucose homeostasis, as indicated by its glucose-lowering and insulin-sensitizing effects. The exact mechanisms involved in the effect of chrysin on insulin sensitivity and hyperglycemia are unclear. However, the effects may be mediated through the amelioration of adipokines, manifested as lower TNF-α and leptin and higher adiponectin levels compared with the untreated rats. These effects may be mediated through the action of chrysin on the adipose tissues (white and brown); however, this suggestion needs further study of the adipose tissues at the molecular and histological levels.

Further, the anti-obesity effect of chrysin might be partially mediated via its lipotropic action, as it significantly corrects dyslipidemia and the hepatic triglyceride content in a dose-dependent manner. This effect may be mediated through the downregulation of the hepatic expression of SREBP-1C compared with the untreated obese rats in a dose-dependent manner. SREBP-1c could be a suitable therapeutic target for obesity and metabolic-associated fatty liver disease [42]. Pai et al. reported that chrysin led to a marked improvement in liver steatosis, and they attributed this improvement to a decrease in SREBP-1c expression and an increase in PPAR-α expression, which resulted in a significant reduction in free fatty acids, triglycerides, and cholesterol [20]. Another mechanism of the anti-obesity, anti-steatosis, and lipotropic effects of chrysin may involve its central action on the apatite, which may affect the caloric and food intake, but in the present study, we did not assess the daily caloric and food intake of rats, so this suggestion needs further investigation.

Several studies revealed that different strategies for NAFLD treatment were related to the activation of the AMPK/mTOR/SREBP1c signaling pathway [43,44,45,46]. In the present study, chrysin treatment showed dose-dependent amelioration in this pathway, with the best effects obtained at a dose of 75 mg/Kg, which effectively normalized the hepatic levels of AMPK, LKB1, and mTOR and their expressions. From all data, we inferred that the LKB1/AMPK/mTOR axis may be a key target for decreasing SREBP1c and ameliorating intrahepatic lipid accumulation and steatosis in obese rats.

Our results agree with Pai et al., who demonstrated the anti-obesity effects of chrysin via the reduction in appetite and suppression of adipocyte hypertrophy and inflammation in adipose tissue 1. Chrysin was shown to improve glucose and lipid metabolism disorders through increasing glycogen synthesis and lipid oxidation and decreasing gluconeogenesis and lipid synthesis in IR HepG2 cells as well as HFD/streptozotocin-induced C57BL/6J mice [21]. In addition, we found that the treatment with chrysin lowered the elevated levels of liver enzymes and bilirubin compared with untreated obese rats. These results concur with other studies regarding the beneficial effect of chrysin against dyslipidemia and hepatic steatosis [47].

The chrysin treatment resulted in a dose-dependent increase in expressions of genes regulating mitochondrial biogenesis, including PGC-1α, NRF-1, and Tfam as well as their protein levels in the livers of obese rats, with the best effects observed in the rats treated with 75 mg/Kg chrysin. Consistently with the current findings, multiple lines of evidence have suggested that the flavonoids belonging to almost all classes promote mitochondrial biogenesis in experimental models. Most of these studies revealed that the upregulation of PGC-1α was a central phenomenon in these processes [48]. Previously, it was reported that adiponectin could regulate mitochondrial biogenesis, fatty acid oxidation and hepatic lipogenesis by signaling through LKB1/AMPK [49]. Moreover, AMPK was shown to activate PGC-1α by phosphorylation or indirectly by activating SIRT1 [50,51]. The activation of AMPK suppresses SREBP-1c by inhibiting mTORC1 and liver X receptor alpha (LXR α) [52]. LKB1 is an upstream kinase activating AMPK by its phosphorylation, and the LKB1/AMPK signaling pathway can regulate hepatic lipogenesis. Previous work in OLETF rats overexpressing AMPKα1 observed a reduction in liver lipogenic gene expression concomitantly with reduced liver steatosis [53].

The above-mentioned chrysin-induced hepatic and metabolic improvements in obese rats were associated with significant improvements in the redox status of the liver. Treatment with chrysin showed a dose-dependent decline in hepatic lipid peroxidation and augmentation of total and reduce GSH and redox ratio. The antioxidant effects of chrysin agreed with previous reports [54,55]. Treatment with chrysin decreased lipid peroxidation products, such as MDA, and increased the activities of free-radical scavenging enzymes and the level of non-enzymatic antioxidant reduced GSH. This mechanism is considered to be due to the capability of the hydroxyl groups located on the fifth and seventh positions in the chrysin to be potent free-radical scavengers. The antioxidant potential may be implicated in the anti-obesity/anti-steatosis effects of chrysin. The antioxidant potential may be implicated in the anti-obesity/anti-steatosis effects of chrysin. The ability of chrysin to enhance the levels of antioxidants is potentially useful in counteracting free-radical-mediated tissue damage caused by hepatotoxicity [55]. The restoration of the GSH system by chrysin plays an important role in anti-obesity treatment because GSH is an essential hepatic antioxidant that controls redox status and plays a key role in restoring insulin sensitivity in obesity-associated metabolic syndrome [56]. Additionally, the elevation of the redox ratio of the hepatocytes indicates high concentrations of thiols, which are associated with a reducing environment in the liver, leading to increased proliferation and differentiation and inhibited cell death. Furthermore, the reducing environment may ameliorate the inflammatory status that may play an important role in alleviating the insulin resistance and the metabolic stress in the obese animals. However, the relationship between chrysin’s antioxidant effects and its anti-obesity or anti-steatosis effects is not well understood and requires further research.

## 5. Conclusions

In conclusion, chrysin has anti-obesity and anti-steatosis effects in HFD-obese rats. Chrysin could mediate its effects by targeting the AMPK/mTOR/SREBP-1c signaling pathway that may mediate its multiple actions, including antioxidant enhancing, glucose and lipid-lowering, metabolic enhancing, mitochondrial boosting, and anti-inflammatory effects.

## Figures and Tables

**Figure 1 molecules-28-01734-f001:**
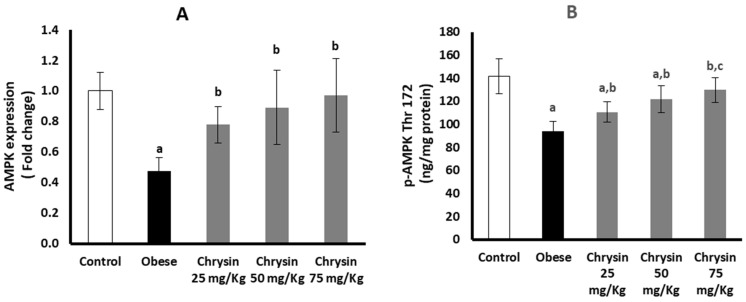
Changes in the hepatic expression of AMPK at mRNA (**A**) and phosphorylated protein (**B**) levels in the obese rats untreated or treated with different doses of chrysin. Data presented as Mean ± SD. a, significantly different from the control group; b, significantly different from the obese group; and c, significantly different from the group treated with chrysin 25 mg/kg, using ANOVA and *p* < 0.05.

**Figure 2 molecules-28-01734-f002:**
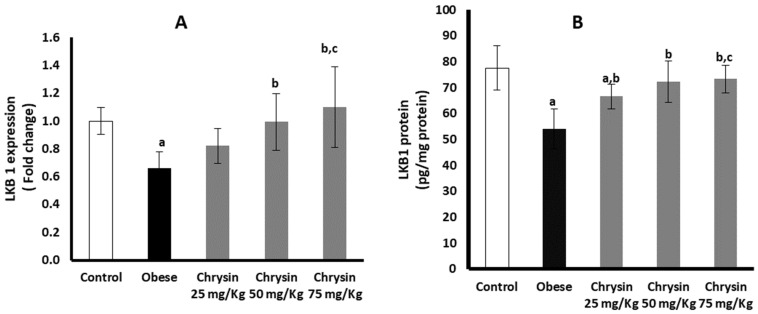
Changes in the hepatic expression of LKB1 at mRNA (**A**) and protein (**B**) levels in the obese rats untreated or treated with different doses of chrysin. Data presented as Mean ± SD. a, significantly different from the control group; b, significantly different from the obese group; c, significantly different from the group treated with chrysin 25 mg/kg, using ANOVA and *p* < 0.05.

**Figure 3 molecules-28-01734-f003:**
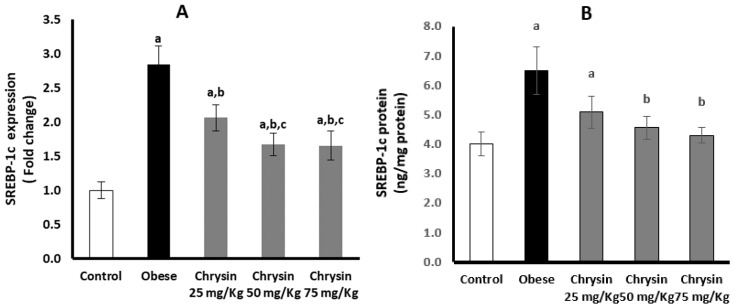
Changes in the hepatic expression of SREBP-1c at mRNA (**A**) and protein (**B**) levels in the obese rats untreated or treated with different doses of chrysin. Data presented as Mean ± SD. a, significantly different from the control group; b, significantly different from the obese group; c, significantly different from the group treated with chrysin 25 mg/kg, using ANOVA and *p* < 0.05.

**Figure 4 molecules-28-01734-f004:**
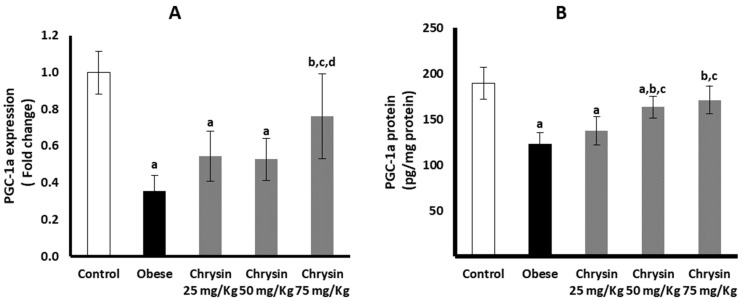
Changes in the hepatic expression of PGC-1α at mRNA (**A**) and protein (**B**) levels in the obese rats untreated or treated with different doses of chrysin. Data presented as Mean ± SD. a, significantly different from the control group; b, significantly different from the obese group; c, significantly different from the group treated with chrysin 25 mg/kg; d, significantly different from the group treated with chrysin 50 mg/kg, using ANOVA and *p* < 0.05.

**Figure 5 molecules-28-01734-f005:**
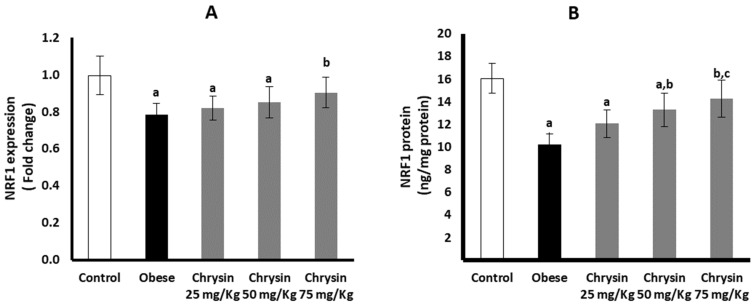
Changes in the hepatic expression of NRF1 at mRNA (**A**) and protein (**B**) levels in the obese rats untreated or treated with different doses of Chrysin. Data presented as Mean ± SD. a, significantly different from the control group; b, significantly different from the obese group; c, significantly different from the group treated with Chrysin 25 mg/kg, using ANOVA and *p* < 0.05.

**Figure 6 molecules-28-01734-f006:**
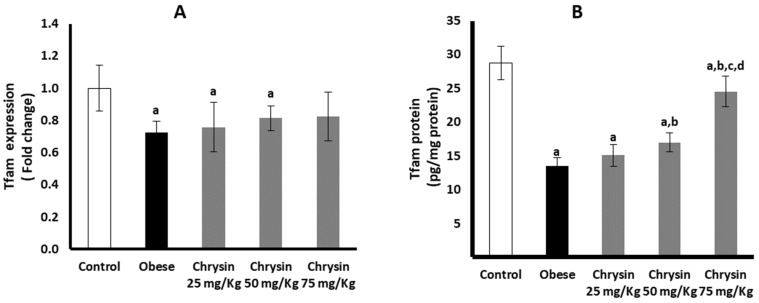
Changes in the hepatic expression of Tfam at mRNA (**A**) and protein (**B**) levels in the obese rats untreated or treated with different doses of chrysin. Data presented as Mean ± SD. a, significantly different from the control group; b, significantly different from the obese group; c, significantly different from the group treated with chrysin 25 mg/kg; d, significantly different from the group treated with chrysin 50 mg/kg, using ANOVA and *p* < 0.05.

**Figure 7 molecules-28-01734-f007:**
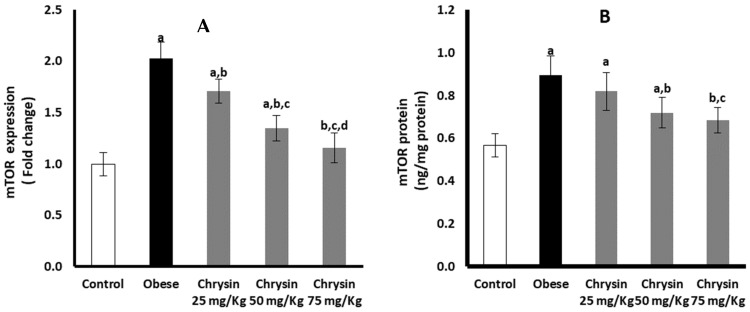
Changes in the hepatic expression of mTOR at mRNA (**A**) and protein (**B**) levels in the obese rats untreated or treated with different doses of chrysin. Data presented as Mean ± SD. a, significantly different from the control group; b, significantly different from the obese group; c, significantly different from the group treated with chrysin 25 mg/kg; d, significantly different from the group treated with chrysin 50 mg/kg, using ANOVA and *p* < 0.05.

**Figure 8 molecules-28-01734-f008:**
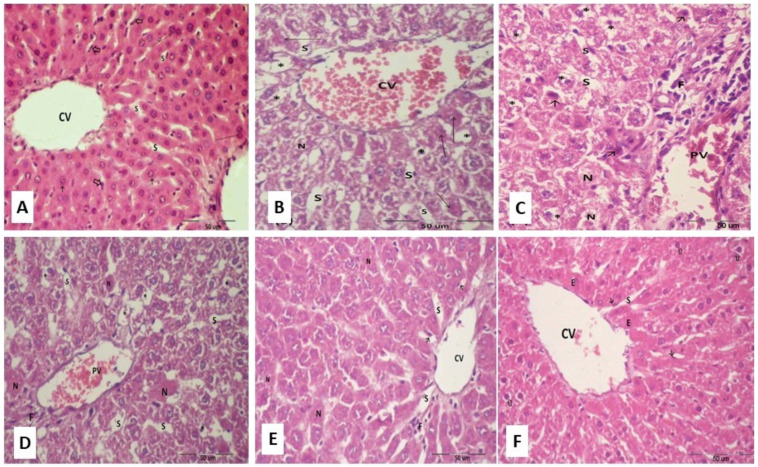
Photomicrographs of liver sections stained with hematoxylin and eosin stain. (**A**) Section of normal control rat showing normal hepatocytes with central vein (CV), sinusoids (S) and Kupffer cells (⇧). Note mononucleated (↑) and binucleated hepatocytes (thin arrow). (**B**,**C**) Liver sections of obese rat showing fatty degeneration (steatosis); where hepatocytes have ballooned clear cytoplasm (*), wide sinusoids (S), apoptotic figures (↑), necrosis (N), and increased eosinophilia (E) in some hepatocytes. (**D**–**F**) Liver section of obese rat after chrysin treatments with increasing doses (25, 50, and 75 mg/Kg/day, respectively) showing sinusoidal dilatation (S) around central vein (CV) with minimal congestion in sinusoids (thin arrow) and well-defined Kupffer cells and low numbers of fatty hepatocytes and apoptotic figures.

**Table 1 molecules-28-01734-t001:** Initial and final body weights, and weight gain of control rats and obese rats untreated and treated with different doses of chrysin.

	Control Rats	Obese Rats			
		Untreated	Chrysin-Treated		
			25 mg/Kg	50 mg/Kg	75 mg/Kg
Initial weight (g)	206.3 ±13.4	388.2 ± 27.9 ^a^	372.6 ± 32.8 ^a^	367.3 ± 27.9 ^a^	365.2 ± 23.1 ^a^
Final weight (g)	232.6 ± 15.5	456.1 ± 35.5 ^a^	427.1 ± 36.0 ^a^	410.7 ± 30.9 ^a,b^	399.9 ± 24.3 ^a,b^
Weight gain (g)	26.3 ± 2.6	67.9 ± 8.7 ^a^	54.6 ± 6.8 ^a,b^	43.4 ± 5.9 ^a,b,c^	34.7± 2.6 ^b,c,d^

Values are presented as (mean ± SD). ^a^ Significantly different from the control group; ^b^ significantly different from the obese group; ^c^ significantly different from the group treated with chrysin 25 mg/kg; ^d^ significantly different from the group treated with chrysin 50 mg/kg, using ANOVA (LSD), *p* value < 0.05.

**Table 2 molecules-28-01734-t002:** Glucose homeostasis parameters, lipid profile, liver function tests in control rats, obese untreated rates, and obese rats treated with different doses of chrysin.

	Control Rats	Obese Rats			
		Untreated	Chrysin-Treated		
			25 mg/Kg	50 mg/Kg	75 mg/Kg
FBS (mg/dL)	86.7 ± 7.4	184.0 ± 12.7 ^a^	156.1 ± 10.6 ^a,b^	137.4 ± 11.3 ^a,b,c^	128.7 ± 8.9 ^a,b,c^
Insulin (mIU/mL)	6.4 ± 1.8	13.3 ± 1.4 ^a^	11.9 ± 0.9 ^a^	9.7 ± 0.8 ^a,b,c^	9.6 ± 0.8 ^a,b,c^
HOMA-IR	1.3 ± 0.3	6.0 ± 0.5 ^a^	4.6 ± 0.3 ^a,b^	3.3 ± 0.3 ^a,b,c^	3.1 ± 0.3 ^a,b,c^
TG (mg/dL)	45.8 ± 4.2	70.5 ± 12.9 ^a^	58.4 ± 4.3 ^a,b^	51.5 ± 3.6 ^b^	49.99 ± 6.1 ^b^
TC (mg/dL)	156.4 ± 11.1	196.8 ± 12.6 ^a^	170.9 ± 10.6 ^b^	161.6 ± 11.3 ^b^	157.7 ± 12.3 ^b^
HDL-c (mg/dL)	47.0 ± 4.7	35.4 ± 2.8 ^a^	39.2 ± 3.1 ^a^	42.5 ± 4.2 ^b^	44.3 ± 3.7 ^b^
LDL-c (mg/dL)	100.0 ± 7.8	147.3 ± 12.2 ^a^	120.0 ± 10.0 ^a,b^	108.8 ± 11.0 ^b^	103.4 ± 11.2 ^b,c^
ALT (U/L)	37.3 ± 4.9	75.1 ± 16.7 ^a^	58.6 ± 6.0 ^a,b^	40.4 ± 5.6 ^b,c^	39.3± 4.3 ^b,c^
AST (U/L)	124.5 ± 21.2	190.1 ± 23.7 ^a^	168.6 ± 22.4 ^a^	142.8 ± 9.9 ^b^	141.0 ± 10.4 ^b,c^
Bil (mg/dL)	0.44 ± 0.06	1.12 ± 0.15 ^a,c^	0.86 ± 0.11 ^a,b^	0.54 ± 0.07 ^b,c^	0.53 ± 0.07 ^b,c^
Hepatic TG (mg/g tissues)	33.7 ± 3.5	99.3 ± 9.1 ^a^	75.2 ± 6.3 ^a,b^	55.4 ± 5.8 ^a,b,c^	53.5 ± 5.4 ^a,b,c^

Values are presented as (mean ± SD) ^a^ Significantly different from the control group; ^b^ significantly different from the obese group; ^c^ significantly different from the group treated with chrysin 25 mg/kg, using ANOVA (LSD), *p* value < 0.05. Abbreviations: ALT, alanine aminotransferase; AST, aspartate aminotransferase; Bil, bilirubin; TG, triglycerides; TC, total cholesterol; FBS, fasting blood sugar, HOMA-IR, homeostasis model assessment index for insulin resistance.

**Table 3 molecules-28-01734-t003:** Adipocytokines levels in control rats, obese untreated rates, and obese rats treated with different doses of chrysin.

	Control Rats	Obese Rats			
		Untreated	Chrysin-Treated		
			25 mg/Kg	50 mg/Kg	75 mg/Kg
TNF-α (pg/mL)	5.5 ± 1.0	63.2 ± 6.7 ^a^	47.4 ± 6.9 ^a,b^	41.8 ± 6.0 ^a,b^	32.9 ± 5 ^a,b,c,d^
Adiponectin (pg/mL)	81.5 ± 7.9	36.4 ± 6.5 ^a^	47.1 ± 5.8 ^a,b^	74.3 ± 6.2 ^b,c^	77.8 ± 6.1 ^b,c^
Leptin (pg/mL)	6.7 ± 0.5	14.5± 1.1 ^a^	11.6 ± 1.0 ^a,b^	9.6 ± 0.8 ^a,b,c^	9.0 ± 0.9 ^a,b,c^

Values are presented as (mean ± SD). ^a^ Significantly different from the control group; ^b^ significantly different from the obese group; ^c^ significantly different from the group treated with chrysin 25 mg/kg; ^d^ significantly different from the group treated with chrysin 50 mg/kg, using ANOVA (LSD), *p* value < 0.05.

**Table 4 molecules-28-01734-t004:** Hepatic MDA, total GSH, GSSG, reduced GSH, and GSH/GSSG ratio in control rats and obese rats untreated or treated with different doses of chrysin.

	Control Rats	Obese Rats			
		Untreated	Chrysin-Treated		
			25 mg/Kg	50 mg/Kg	75 mg/Kg
MDA (nmol/g tissues)	28.1 ± 2.9	71.7 ± 5.6 ^a^	51.8 ± 4.3 ^a,b^	43.6 ± 3.8 ^a,b,c^	42.6 ± 4.0 ^a,b,c^
tGSH (nmol/mg protein)	27.7 ± 1.7	21.4 ± 1.8 ^a^	23.7 ± 1.9 ^a^	24.7 ± 1.9 ^a,b^	25.0 ± 1.7 ^a,b^
GSSG (nmol/mg protein)	1.1 ± 0.1	2.0 ± 0.2 ^a^	1.5 ± 0.2 ^a,b^	1.2 ± 0.2 ^b,c^	1.2 ± 0.1 ^b,c^
GSH (nmol/mg protein)	25.6 ± 1.6	17.5 ± 1.5 ^a^	20.8 ± 1.7 ^a,b^	22.3 ± 1.8 ^a,b^	22.7 ± 1.6 ^a,b^
GSH/GSSG ratio	23.6 ± 2.0	8.9 ± 0.6 ^a^	14.1 ± 1.2 ^a,b^	19.0 ± 2.5 ^a,b,c^	19.7 ± 1.9 ^a,b,c^

Values are presented as (mean ± SD). ^a^ Significantly different from the control group; ^b^ significantly different from the obese group; ^c^ significantly different from the group treated with chrysin 25 mg/kg, using ANOVA (LSD), *p* value < 0.05. MDA, malondialdehyde; tGSH, total glutathione; GSSG, glutathione disulphide; GSH, reduced glutathione.

**Table 5 molecules-28-01734-t005:** Primers were used for the PCR amplification.

Gene	Accession Number	Primer Sequence	
18S rRNA	NR_046237.2	F:	GTAACCCGTTGAACCCCATT
		R:	CAAGCTTATGACCCGCACTT
PGC-1α	NM_031347.1	F:	GTGCAGCCAAGACTCTGTATGG
		R:	GTCCAGGTCATTCACATCAAGTTC
NRF-1	NM_001100708.1	F:	TTACTCTGCTGTGGCTGATGG
		R:	CCTCTGATGCTTGCGTCGTCT
Tfam	NM_031326.2	F:	CCCACAGAGAACAGAAACAG
		R:	CCCTGGAAGCTTTCAGATACG
AMPK	NM_023991.1	F:	GTGGTGTTATCCTGTATGCCCTTCT
		R:	CTGTTTAAACCATTCATGCTCTCGT
LKB1	NM_001108069.2	F:	GAGGAAGTGGGTCAGAATGGA
		R:	CCGGCCTTCTGGCTTCA
SREB1C	NM_001276708.1	F:	GACGACGGAGCCATGGATT
		R:	GGGAAGTCACTGTCTTGGTTGTT
mTOR	NM_019906.2	F:	TTGGAGTGGCTGGGTGCTGA
		R:	AAGGGCTGAACTTGCTGGAA

Where F: Forward primer and, R: Reverse primer.

## Data Availability

All data available upon your request to the corresponding author.

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
