# Peer review of "The Anti-Obesity and Anti-Steatotic Effects of Chrysin in a Rat Model of Obesity Mediated through Modulating the Hepatic AMPK/mTOR/lipogenesis Pathways"

_molecules, 2023, doi:10.3390/molecules28041734_

Round 1

Reviewer 1 Report

Abstract:

-Did you use male or female rats?

-The method part must be written in a more clear pattern. How did you induce obesity?

- Replace "improve" with another clear word like decrease, increase,...

-"oxidative stress"?? what markers exactly?

- Choose keywords according to MeSH.

Introduction:

- Avoid using unnecessary abbreviations.

- Introduction does not follow a logic pattern.

- There are some unnecessary information. But, some important information is missing especially about chrysin.

- The aim of the study is not clear enough.

Materials and methods:

- Write materials in a paragraph.

- Why did you use male rats?

- There are some typing and grammatical mistakes throughout the text.

- "20 heavier"?? 20 g??

-How did you choose the concentrations, route of administration of chrysin? Also, the length of study?

- Why there is no chrysin alone group?

- why there is no positive control group?

- Introduce the full form of abbreviations the first time you use them.

- "tumor necrosis factor-alpha (TNFα)" is wrong. TNF-α is correct.

- The dose of Ketamine and Xylazine?

- This part is not written very organized.

- 2.8, 2.9, and 2.10 must be written more completely.

Results

- write P values in the text. Compare untreated obese group with the control groups. and Chrysin group with the untreated obese group.

Discussion

This part must be rewritten to follow a logical pattern.

Reviewer 2 Report

The authors investigated the involvement of Chrysin in rat obesity model. Overall, the study is superficial, and thus will have limited impact on the field. This article is novel and original (J Tradit Complement Med . 2019 Sep 6;10(6):577-585. Chrysin mitigated obesity by regulating energy intake and expenditure in rats) is insufficient for publication in the Molecules.

1.There are errors in the introduction and discussion, and many statements are cited as documents that are not representative. You should use representative articles or high-impact-factor reviews, not random articles from random journals. References are available, but the introduction or discussion sections lack much description of the use of Chrysin.

2. Please explain why you used the concentration 25~75 mg/kg, and its relevant biological significance, especially when you claim its potential candidate.

3. The abstract needs to be revised. The parameters used to evaluate the study objectives were not indicated.

4. How animals are sacrificed, perfused, and brains were isolated.

5. In Discussion: The reason for using a Chrysin requires in-depth discussion.

6. An evaluation of histopathological effects on adipose tissue should be presented.

Round 2

Reviewer 1 Report

The manuscript has been improved. But there are still some writing mistakes. The abbreviations usage must be checked, too.
